# High-risk fertility behaviors and associated factors among married reproductive-age women in sub-Saharan Africa: A multilevel mixed-effect analysis of nationally representative data from 35 countries

**Kusse Urmale Mare**[1]*, **Setognal Birara Aychiluhm**[2,3], **Getahun Fentaw Mulaw**[4,5], **Kebede Gemeda Sabo**[1], **Mekuriyaw Gashaw Asmare**[1], **Betel Zelalem Wubshet**[1], **Tsion Mulat Tebeje**[6], **Beminate Lemma Seifu**[7]

1 Department of Nursing, College of Medicine and Health Sciences, Samara University, Samara, Ethiopia, 2 Department of Epidemiology & Biostatistics, Institute of Public Health, College of Medicine & Health Sciences, University of Gondar, Gondar, Ethiopia, 3 Rural Health Research Institute, Charles Sturt University, Orange, NSW, Australia, 4 School of Public Health, College of Medicine and Health Sciences, Woldia University, Woldia, Ethiopia, 5 School of Pharmacy and Medical Sciences, Griffith University, Gold Coast, QLD, Australia, 6 School of Public Health, College of Health Sciences and Medicine, Dilla University, Dilla, Ethiopia, 7 Department of Public Health, College of Medicine and Health Sciences, Samara University, Samara, Ethiopia

* kussesinbo@gmail.com, hulumlebego2019@gmail.com

**Data Availability Statement:** The raw dataset used and analyzed in this study can be openly accessed

## Abstract

### Background

Although high-risk fertility behaviors are linked with poor maternal and child health outcomes, their prevalence remains higher in resource-limited countries and varies significantly by context. Evidence on the recent estimates of these fertility risks at the sub-Saharan Africa level is limited. Therefore, this study aimed to examine the pooled prevalence of high-risk fertility behaviors and associated factors among married women in this region.

### Methods

Data from DHS of 35 sub-Saharan African countries were used and a weighted sample of 243,657 married reproductive-age women were included in the analysis. A multilevel binary logistic regression models were fitted and the final model was selected based on the log-likelihood and deviance values. A p-value less than 0.05 and an adjusted odds ratio with a corresponding 95% confidence interval were used to identify the factors associated with high-risk fertility behaviors.

### Results

The pooled prevalence of high-risk fertility behaviors among women in sub-Saharan Africa was 77.7% [95% CI = 77.6%-77.9], where 43.1% [95% CI: 42.9%-43.3%], and 31.4% [95% CI = 31.2%-31.6%] had a single risk and combination of two or three fertility risks,

from the DHS website (https://dhsprogram.com/data/download/urlslist_178265.txt).

**Funding:** The authors received no specific funding for this work.

**Competing interests:** The authors have declared that no competing interest exist.

respectively. The highest level of single-risk fertility pattern was observed in Burundi (53.4%) and Chad had the highest prevalence of both at least one (89.9%) and multiple (53.6%) fertility risks. Early and polygamous marriages, low maternal and husband education, poor wealth index, unmet need for contraception, couple's fertility discordance, rural residence, high community-level early marriage practice, and low community-level women empowerment were associated with risky fertility behaviors.

## Conclusions

More than three-quarters of married women in SSA were engaged in high-risk fertility behaviors, with significant variations across the included countries. Therefore, addressing the modifiable risk factors like improving access to need-based contraceptive methods and empowering couples through education for a better understanding of their reproductive health with particular attention to rural settings are important in reducing these fertility risks. The results also suggest the need to strengthen the policies regulating the prohibition of early and polygamous marriages.

## Background

High-risk fertility behaviors are maternal reproductive characteristics that include childbirth at a younger age (<18 years) or advanced age (>34 years), suboptimal birth spacing (<24 months), and birth order of 4 or more [1]. Studies have shown that these fertility behaviors are associated with maternal and child morbidity and mortality [2–5]. Pregnancy and birth at extreme ages, shorter birth intervals, and high parity result in an increased risk of maternal undernutrition [6, 7], anemia [6], pregnancy and birth complications [8, 9], and mortality [10]. Similarly, maternal exposure to these fertility risks has been linked to neonatal, infant, and under-five child mortality [4, 11, 12], stillbirths [4, 11], low birth weight and preterm births [4, 5, 9, 13], malnutrition [3, 13, 14], and other comorbidities [11, 14].

Existing evidence shows that children of mothers with high-risk fertility behaviors had a higher risk of death [3, 8, 15], with an elevated mortality risk for children of mothers with multiple-risk behaviors [3]. For instance, compared to children of mothers who did not experience high-risk fertility behaviors, children of mothers with a single fertility risk had an added mortality risk of 7%, and the death probability increased by 39% among children of mothers with multiple fertility risks [8]. Mortalities among children born to mothers with risky fertility behaviors were associated with increased occurrence of poor perinatal outcomes like low birth weight and preterm birth [13, 16–18], low Apgar score [16–18], and stillbirth and congenital anomaly [12]. Furthermore, maternal high-risk fertility behaviors were also linked with childhood illnesses like acute respiratory infections, diarrhea, fever [11], and undernutrition [3, 14].

In addition to its short- and long-term consequences on the well-being of the child, risky fertility behaviors have also been linked to maternal mortality [10] by increasing maternal exposure to poor obstetric outcomes, nutritional deficiencies, and medical illness. Accordingly, the risk of preeclampsia, puerperal endometritis, and systemic infection was higher among mothers who experienced risky fertility behaviors [19]. Besides, mothers in the high-risk fertility category had increased odds of chronic malnutrition [6, 7] and anemia [6].

Approximately 21 million births occur to adolescent mothers (11–19 years of age) in resource-limited settings, of which 777,000 occur among younger adolescents under 15 years

of age [20]. Adolescent pregnancy also ranges from 7.2% to 44.3% in Sub-Saharan Africa (SSA) [21] and 9.2% to 21.5% in Africa [22]. Studies have linked early childbearing as the cause of higher-order births or high fertility, which is associated with poor maternal and neonatal outcomes. According to recent estimates, the total fertility rate for women in SSA was 4.7 [23] and the highest prevalence of short birth interval was reported in Chad (30.2%), Congo (27.1%) [24, 25], and Ethiopia 47% [26]. Furthermore, in Eastern Africa, the magnitude of high-risk fertility behaviors among women ranged from 41% in Zimbabwe to 66.6% in Uganda [27].

Different socio-demographic and obstetric characteristics were reported to have a significant influence on the maternal tendency to experience high-risk fertility behaviors. In this regard, maternal and husband education [27–30], place of residence [27, 31, 32], household wealth [27, 28, 32], access to health facilities [27, 33], use of maternal health care services (like antenatal and skilled delivery care, and contraceptives) [27, 31, 34], history of adverse birth outcomes [27, 28], and breastfeeding practices [34, 35] were the determinants of high-risk fertility behaviors.

Despite the existence of cost-effective interventions (i.e. family planning programs) that simultaneously prevent all three forms of fertility risks, the level of risky fertility behaviors in African regions is relatively high [27, 31, 36]. Moreover, recent evidence on the magnitude and determinants of these fertility patterns at the SSA level is scarce. A previous study that was conducted in this region was restricted to 27 countries in SSA [36] and other studies were done at a single country [31, 33, 37] and region (Eastern Africa) [27] levels and some did not assess the effect of some community-level variables [27, 31, 33]. Evidence of the contextual factors underlying high-risk fertility behaviors is crucial in designing context-specific interventions targeted to address risky fertility behaviors among women in the region. Therefore, this study aimed to examine the pooled prevalence and determinants of high-risk fertility behaviors among women of reproductive age in SSA.

## Methods

### Study settings and data source

The study used data from the recent Demographic and Health Surveys (DHS) conducted in 35 SSA countries between 2010 and 2021. The selection of countries was based on the availability of a standardized and unrestricted DHS dataset that contains the outcome and necessary explanatory variables (Fig 1). The DHS is a nationally representative survey conducted regularly every five years to collect data on basic sociodemographic characteristics and various health indicators. The surveys across all countries used a standardized methodology and the same two-stage stratified cluster sampling technique to select the study participants. In the first stage, Enumeration Areas (EAs) were randomly selected based on the country's recent population, and using the housing census as a sampling frame, households were randomly selected in the second stage. For this study, a total weighted sample of 243,657 married women of reproductive age was considered for the analysis. Nulliparous and primiparous women and those who had missing data on the three indices of HRFB (age at childbirth, birth interval, and birth order) were excluded from the analysis. Detailed information about DHS methodology can be accessed online (https://dhsprogram.com/Methodology/index.cfm).

### Study variables and definitions

**Outcome variable.** The outcome variable for this study was "high risk-fertility behavior" that was measured using three fertility indicators: maternal age $< 18$ or $> 34$ years at the birth of the index child, duration of preceding birth interval of less than 24 months, and birth order

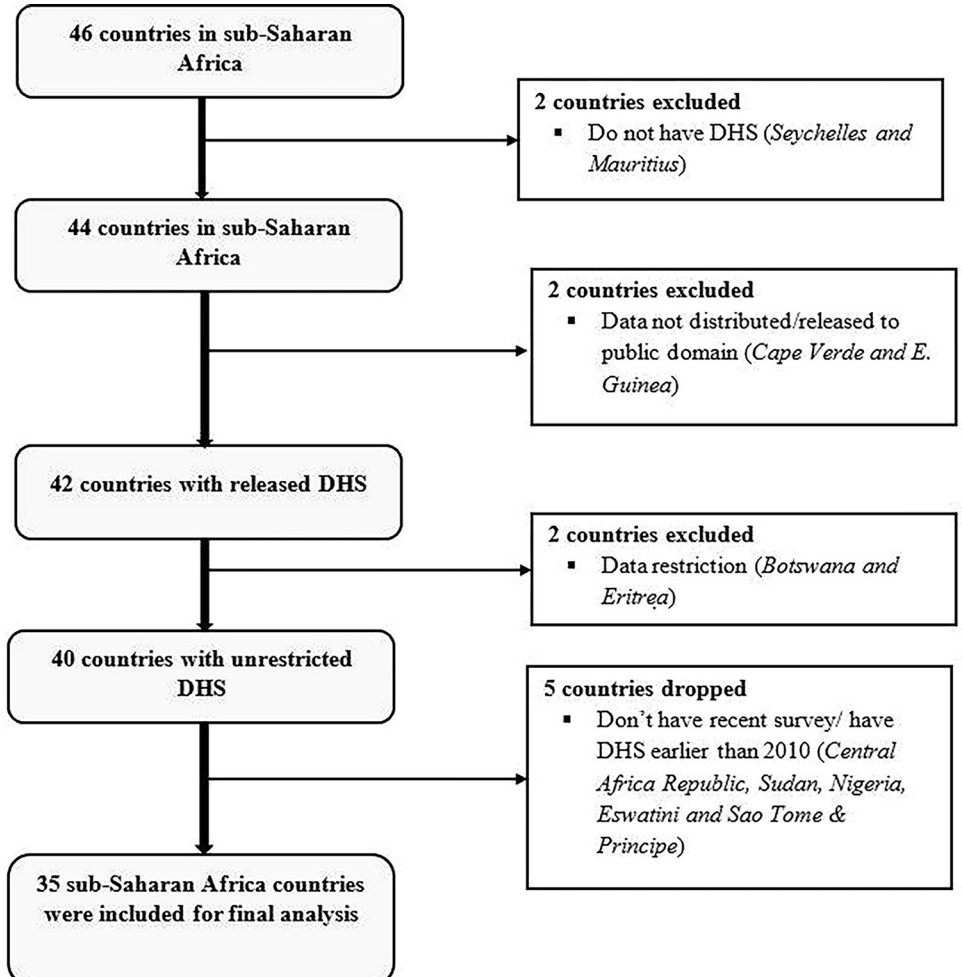

**Fig 1. Flow scheme for inclusion and exclusion of sub-Saharan African countries in this study.**

of four or above. Thus, a woman with any one of the three indicators was considered as having high-risk fertility behavior (coded as "1") and otherwise labeled as not having risky fertility behavior (coded as "0"). Furthermore, for descriptive purposes, this variable was recategorized as "single HRFB" for women with only one risky fertility behavior and "multiple HRFB" for women with a combination of two or three risks.

**Explanatory variables.**   Individual-level variables included current age, age at marriage, type of marriage, woman's and partner's education, women's employment status, media exposure, household head, wealth index, women's participation in the household decisions, knowledge of ovulatory cycle, previous contraceptive use, unmet need for contraception, couple's fertility preference, and history of pregnancy loss. While, place of residence, SSA regions, community-level media exposure, community-level women illiteracy, community-level women empowerment, community-level early marriage practice, and community-level husband literacy were considered as the community-level variables. The selection of these variables was based on the literature [7, 27, 31, 33, 36, 38]

Exposure to mass media was computed using three variables (frequency of watching television, listening to the radio, and reading newspapers), for which there are three response options (i.e., not at all, less than once a week, and at least once a week). Therefore, women who

reported watching television or listening to the radio, or reading the newspaper less than once a week and at least once a week were considered as having media exposure (coded = Yes "1"), while those who reported not watching television or listening to the radio or reading a newspaper at all were labeled as not having exposure to mass media (coded = No "0").

Couple's fertility preference: women who reported that their husbands preferred to have the same number of children were regarded as having "concordant fertility preference", while those whose partners desired to have fewer or more children than their desire were considered as having "discordant fertility preference".

Women empowerment was generated by computing four variables (i.e. who usually decides on women's earnings, health care, large household purchase, and a family visit) that have four responses (respondent alone, respondent and partner, partner alone, and someone else). Thus, women were considered to have been empowered if they reported that the decision was made by themselves or jointly with their partner and otherwise regarded as not empowered.

Other community-level variables (i.e., community-level media exposure, women's illiteracy and empowerment, early marriage practice, and husband's literacy) were generated by aggregating the individual-level observations at the cluster level and the aggregates were computed using the average values of the proportions of women in each category of a given variable and median values were used to categorize the aggregated variables into two groups as "low" and "high".

**Data management and statistical analysis.** Stata version 17 was used for data cleaning and analysis. Before analysis, the availability of the outcome variable in the DHS dataset of each country was confirmed and all variables included in the study were checked for missing values. Then, the datasets of 35 SSA countries were appended and weighted to compensate for the non-representativeness of the sample (unequal probability of selection and response rate) and obtain reliable estimates and standard errors.

To account for the clustering effects (i.e. women are nested within clusters), a multilevel logistic regression analysis was used to determine the effects of each predictor on maternal high-risk fertility behaviors. Bivariable multilevel logistic regression analysis was done and all variables with a p-value of less than 0.25 in this analysis were included in the multivariable multilevel logistic regression analysis [39, 40].

In our analysis, we fitted four models to select the model that best fits the data: Model I (a model without independent variables to test random variability in the intercept), Model II (a model with only individual-level explanatory variables), Model III (a model with only community-level explanatory variables), and Model IV (a model with both individual and community-level predictors). Then, log-likelihood (LL) and deviance (i.e. -2*LL) values were used for model selection and the model with the lowest deviance and highest LL values was considered as a best-fitted model for the final analysis. The presence of multi-collinearity among the explanatory variables was checked using a generalized variance inflation factor value and this value was less than 10 for all variables, suggesting that there was no multi-collinearity. Finally, in the multivariable analysis, a p-value less than 0.05 and an adjusted odds ratio with a corresponding 95% confidence interval were used to identify the factors associated with high-risk fertility behaviors. In addition, variation in the prevalence of high-risk fertility behavior across clusters was estimated using the intra-class correlation coefficient (ICC), proportion change in variance (PCV), and median odds ratio (MOR). The manuscript complies with the "Enhancing the Quality and Transparency of Health Research" network guidelines for strengthening the reporting of observational studies.

**Ethical approval.** Permission to access (AuthLetter_178265) the data used in the present study was granted from a measure DHS via an online request at http://www.dhsprogram.com. Prior to usage, data were made anonymized by ensuring that no personal identifications linked

with individual participant were not contained in it. The accessed data was only used for this registered study and are publicly available from the program's official database.

## Results

### Sociodemographic and reproductive characteristics

Of 243,647 women included in the analysis, the majority (87.3%) were over the age of 35, and three-quarters (75.3%) were in monogamous relationships. The majority (42.2%) of participants had no formal education and 65.7% had exposure to media. More than one-third ((38.6%) were from rich households and 56.9% were involved in the household decision-making processes. Almost half (49.3%) of the respondents were married before the age of 18 and 51.2% had ever used contraception. Additionally, 23.6% of women had unmet family planning needs and 66% had differing fertility preferences with their partners.

Regarding the distribution of risky fertility behaviors by background characteristics, women aged 35 to 49 years had the highest prevalence of single (49.9%), multiple (48.4%), and overall (49.2%) risky fertility behaviors compared to women in other groups. It was also found that the proportions of high-risk fertility behaviors among uneducated and non-working women, and those married at younger ages were 46.9%, 33.2%, and 49.3%, respectively (Table 1).

**Prevalence of high-risk fertility behaviors.** The overall prevalence of high-risk fertility behaviors among married reproductive-age women in SSA was 77.7% [95% CI = 77.6%-77.9%], of which 43.1% [95% CI: 42.9%-43.3%] and 31.4% [95% CI = 31.2%-31.6%] had single and multiple risks, respectively. Disparities across the countries were observed in the magnitude of overall, single, and multiple high-risk fertility behaviors. The countries with the highest overall prevalence of risky fertility behaviors were Chad (89.9%), Niger (88%), Mali (83.1%), Ethiopia (82%), and Angola (82%), and the lowest prevalence was found in South Africa (45.3%). In addition, South Africa has the lowest prevalence of both single (34.1%) and multiple high-risk fertility behaviors (11.2%), while Burundi (54.3%) and Chad (53.6%) were the countries where the highest proportion of women experienced single and multiple fertility risks. Details on the prevalence of risky fertility behavior across SSA countries are presented in Figs 2–4.

**Random effect analysis.** The ICC value in the null model indicates that 18% of the variation in high-risk fertility behaviors across the clusters was attributed to community-level factors and the variability was reduced to 13% after considering both individual and community-level variables in the final model. The final model's PCV value of 0.316 implies that the combined influence of individual and community-level variables accounted for 31.6% of the variation in high-risk fertility practices at the community level. Additionally, the presence of heterogeneity in high-risk fertility behaviors between clusters was indicated by the MOR with a value of 2.201. This shows that compared to the clusters with a low proportion of risky fertility behaviors, women in the clusters with a higher prevalence of risky fertility patterns had about 2.2 times higher likelihood of engaging in such behaviors. Model IV had the lowest deviance value (i.e. 225182) and was therefore selected as the best-fitted model (Table 2).

**Factors associated with high-risk fertility behaviors.** It was found that women who had their first marriage before the age of 18 years were about 3.7 times more likely to engage in high-risk pregnancy behavior than those who married at or after the age of 18 years [AOR (95% CI): 3.70 (3.59, 3.81)]. The odds of experiencing risky fertility behaviors among women in polygamous relationships (whose husbands had other wives) were increased by 32% compared to those in monogamous unions [AOR (95% CI): 1.32 (1.28, 1.37)]. This study also revealed that women with primary education [AOR (95% CI): 1.67 (1.61, 1.73)] and no formal

**Table 1.  Distribution of high-risk fertility behaviors across women's socio-demographic and reproductive characteristics in sub-Saharan African countries.**

| Characteristics | High-risk fertility behavior | | | | Total |
|---|---|---|---|---|---|
| | No risk | Single risk | Multiple risks | Overall HRFB | |
| **Individual-level characteristics** | | | | | |
| **Woman's current age** | | | | | |
| 15–24 | 7,587(13.1) | 16,730 (15.9) | 6,740 (8.4) | 23,470 (12.6) | 31,057 (12.7) |
| 25–34 | 35.197 (60.8) | 35,965 (34.2) | 34,838 (43.2) | 70,803 (38.2) | 106,000 (43.5) |
| 35–49 | 15,143 (26.1) | 52,409 (49.9) | 39,047 (48.4) | 91,457 (49.2) | 106,600 (43.8) |
| **Nature of marriage** | | | | | |
| Monogamy | 46,151.2 (83.2) | 77,307 (75.8) | 54,257 (69.1) | 131,564 (72.9) | 177,715 (75.3) |
| Polygamy | 9,347 (16.8) | 24,579 (24.1) | 24,231 (30.9) | 48,810 (27.1) | 58,157 (24.7) |
| **Woman's education** | | | | | |
| No formal education | 15,637 (27.0) | 44,077 (41.9) | 42,999 (53.3) | 87,076 (46.9) | 102,713 (42.2) |
| Primary education | 18,000 (31.1) | 36,725 (34.9) | 27,260 (33.8) | 63,984 (34.4) | 81,985 (33.6) |
| Higher education | 24,290 (41.9) | 24,302 (23.2) | 10,367 (12.9) | 34,669 (18.7) | 58,959 (24.2) |
| **Woman's employment** | | | | | |
| Not working | 18,244 (32.9) | 33,099 (32.5) | 26,761 (34.1) | 59,860 (33.2) | 78,104 (33.1) |
| Working | 37,247 (67.1) | 68,732 (67.5) | 51,661 (65.9) | 120,393 (66.8) | 157,641 (66.9) |
| **Media exposure** | | | | | |
| Yes | 43,240 (74.7) | 68,982 (65.7) | 47,579 (59.1) | 116,516 (62.9) | 159,802 (65.7) |
| No | 14,625 (25.3) | 36,002 (34.3) | 32,910 (40.9) | 68,912 (37.1) | 83,537 (34.3) |
| **Partner education** | | | | | |
| No formal education | 15,374 (27.7) | 39,543 (38.8) | 36,912 (47.1) | 76,455 (42.4) | 91,829 (39.0) |
| Primary education | 14,528 (26.2) | 30,060 (29.5) | 22,861 (29.1) | 52,921 (29.4) | 67,449 (28.6) |
| Higher education | 25.566 (46.1) | 32,235 (31.7) | 18,676 (23.8) | 50,912 (28.2) | 76,477 (32.4) |
| **Head of household** | | | | | |
| Male | 48,014 (82.9) | 98,356 (85.0) | 69,802 (86.6) | 159,159 (85.7) | 207,171 (85.0) |
| Female | 9,913 (17.1) | 15,749 (15.0) | 10,824 (13.4) | 26,573 (14.3) | 36,487 (15.0) |
| **Household wealth index** | | | | | |
| Poor | 17,873 (30.9) | 43,676 (41.6) | 38,684 (48.0) | 82,361 (44.3) | 100,234 (41.1) |
| Middle | 10,258 (17.7) | 21,899 (20.8) | 17,242 (21.4) | 39,140 (21.1) | 49,398 (20.3) |
| Rich | 29,797 (51.4) | 39,529 (37.6) | 24,700 (30.6) | 64,229 (34.6) | 94,026 (38.6) |
| **Woman's empowerment** | | | | | |
| No participation | 26,732 (48.2) | 44,281 (43.5) | 30,612 (39.0) | 74,893 (41.5) | 101,626 (43.1) |
| Has participation | 28,757 (51.8) | 57,577 (56.5) | 47,857 (61.0) | 105,434 (58.5) | 134,192 (56.9) |
| **Age at first cohabitation** | | | | | |
| ≥18 year | 44,635 (77.1) | 60,484 (57.6) | 18,489 (22.9) | 78,973 (42.5) | 123,608 (50.7) |
| <18 year | 13,292 (22.9) | 44,620 (42.4) | 62,137 (77.1) | 106,757 (57.5) | 120,049 (49.3) |
| **Knowledge of the ovulatory cycle** | | | | | |
| Yes | 17,129 (30.9) | 27,733 (27.2) | 20,366 (26.0) | 48,098 (26.7) | 65,228 (27.7) |
| No | 38,383 (69.1) | 74,126 (72.8) | 58,085 (74.0) | 132,212 (73.3) | 170,594 (72.3) |
| **Ever used contraceptives** | | | | | |
| Yes | 34,714 (59.9) | 54,120 (51.5) | 35,803 (44.4) | 89,923 (48.4) | 124,637 (51..2) |
| No | 23,213 (40.1) | 50,989 (48.5) | 44,823 (55.6) | 95,807 (51.6) | 119,020 (48.8) |
| **Unmet need for FP** | | | | | |
| No | 44,761 (80.6) | 76,732 (75.3) | 58,787 (74.9) | 135,520 (75.1) | 180,281 (76.4) |
| Yes | 10,758 (19.4) | 25,162 (24.7) | 19,712 (25.1) | 44,874 (24.9) | 55,632 (23.6) |
| **Couple's fertility preference** | | | | | |
| Concordant | 22,423 (40.9) | 34,114 (34.2) | 22,027 (28.7) | 56,141 (31.8) | 78,565 (34.0) |

*(Continued)*

**Table 1.** (Continued)

| Characteristics | High-risk fertility behavior | | | | Total |
|---|---|---|---|---|---|
| | No risk | Single risk | Multiple risks | Overall HRFB | |
| Discordant | 32,449 (59.1) | 65.725 (65.8) | 54,676 (71.3) | 120,401 (68.2) | 152,850 (66.0) |
| **History of pregnancy loss** | | | | | |
| No | 45,778 (82.5) | 83,923 (82.4) | 64,653 (82.4) | 148,576 (82.3) | 194,353 (82.4) |
| Yes | 9,744 (17.5) | 17,975 (17.6) | 13,847 (17.6) | 31,823 (17.6) | 41,567 (17.6) |
| **Community level characteristics** | | | | | |
| **Residence** | | | | | |
| Urban | 25,303 (43.7) | 33,197 (31.6) | 20,745 (25.7) | 53,942 (29.0) | 79,246 (32.5) |
| Rural | 32,624 (56.3) | 71,907 (68.4) | 59,880 (74.3) | 131,179 (71.0) | 164,412 (67.5) |
| **SSA regions** | | | | | |
| Central Africa | 9,603 (16.6) | 19,812 (18.9) | 15,343 (19.0) | 25,155 (18.9) | 44,757 (18.4) |
| Eastern Africa | 12,114 (20.9) | 20,257 (19.3) | 16,091 (20.0) | 36,348 (19.6) | 48,463 (19.9) |
| Southern Africa | 11.123 (19.2) | 15,842 (15.1) | 10,058 (12.5) | 25,900 (13.9) | 37,023 (15.2) |
| Western Africa | 25,087 (43.3) | 49,193 (46.8) | 39,134 (48.5) | 88,327 (47.6) | 113,414 (46.5) |
| **Community-level media exposure** | | | | | |
| High | 13,801 (23.8) | 22,292 (21.2) | 15,765 (19.6) | 38,057 (20.5) | 51,858 (21.3) |
| Low | 44,126 (76.2) | 82,812 (78.8) | 64,860 (80.4) | 147,673 (79.5) | 191,799 (78.2) |
| **Community-level women illiteracy** | | | | | |
| Low | 6,657 (11.5) | 10,724 (10.2) | 6,980 (8.7) | 17,705 (9.5) | 24,361 (10.0) |
| High | 51,270 (88.5) | 94,380 (89.8) | 73,645 (91.3) | 168,025 (90.5) | 219,296 (90.0) |
| **Community-level women empowerment** | | | | | |
| High | 16,476 (28.4) | 28,695 (72.3) | 21,035 (26.1) | 49,730 (26.8) | 66,206 (27.2) |
| Low | 41,448 (71.6) | 76,404 (72.7) | 59,589 (73.9) | 135,993 (73.2) | 177,441 (72.8) |
| **Community-early marriage practice** | | | | | |
| Low | 21,255 (36.7) | 34,257 (32.6) | 22,764 (28.2) | 57,021 (30.7) | 78,276 (32.1) |
| High | 36,672 (63.3) | 70,847 (67.4) | 57,862 (71.7) | 128,709 (69.3) | 165,381 (67.9) |
| **Community-level male education** | | | | | |
| High | 8,782 (15.2) | 14,379 (13.7) | 9,958 (12.3) | 24,338 (13.1) | 33,120 (13.6) |
| Low | 49,142 (84.8) | 90,719 (86.3) | 70,666 (87.7) | 161,385 (86.9) | 210,527 (86.4) |

HRFB: high-risk fertility behavior; FP: family planning; SSA: sub-Saharan Africa.

education [AOR (95% CI): 2.14 (2.06, 2.23)] were more likely to experience high-risk fertility behavior compared to those who had at least a secondary education. Likewise, women whose husbands had primary education [AOR (95% CI): 1.23 (1.11, 1.27)] and no education [AOR (95% CI): 1.50 (1.44, 1.62)] were more likely to experience risky fertility behavior. Furthermore, being from a household with middle [AOR (95% CI):1.28 (1.23, 1.33)] and poor wealth indexes [AOR (95% CI):1.27 (1.22, 1.32)], having an unmet need for family planning [AOR (95% CI): 1.38 (1.34, 1.43)], and couple's fertility discordance [AOR (95% CI): 1.15 (1.12, 1.19)] were the other individual-level factors affecting high-risk fertility behavior.

Regarding the community-level factors, women living in rural areas [AOR (95% CI): 1.86 (1.81, 1.93)] were about 86% more likely to have high-risk fertility patterns compared to women living in urban residences. Likewise, women from communities with high early marriage practice [AOR (95% CI): 1.37 (1.42, 1.49)] and low community-level women empowerment [AOR (95% CI): 1.24 (1.22, 1.27)] had greater tendency to engage in high-risk fertility behaviors compared to their respective counter groups (Table 3).

| Country | Single HRFB | Total | ES (95% CI) | Weight (%) |
|---|---|---|---|---|
| Angola | 2795.98 | 6,586.05 | 42.45 (41.26, 43.65) | 2.70 |
| Burkina Faso | 5094.428 | 10,658.91 | 47.80 (46.85, 48.74) | 4.28 |
| Benin | 4042.357 | 8,861.71 | 45.62 (44.58, 46.65) | 3.58 |
| Burundi | 4472.999 | 8,240.97 | 54.28 (53.20, 55.35) | 3.32 |
| DR Congo | 3581.982 | 8,124.94 | 44.09 (43.01, 45.17) | 3.30 |
| Congo | 2043.025 | 4,764.03 | 42.88 (41.48, 44.29) | 1.95 |
| Cote Divoire | 1653.0179 | 4,007.57 | 41.25 (39.72, 42.77) | 1.66 |
| Cameroon | 2525.62 | 6,082.02 | 41.53 (40.29, 42.76) | 2.51 |
| Ethiopia | 3048.747 | 7,885.36 | 38.66 (37.59, 39.74) | 3.33 |
| Gabon | 1314.734 | 3,037.15 | 43.29 (41.53, 45.05) | 1.24 |
| Ghana | 1868.986 | 4,185.91 | 44.65 (43.14, 46.16) | 1.70 |
| Gambia | 2539.86 | 5,562.93 | 45.66 (44.35, 46.97) | 2.24 |
| Guinea | 2560.095 | 5,862.19 | 43.67 (42.40, 44.94) | 2.39 |
| Kenya | 6057.492 | 14,574.44 | 41.56 (40.76, 42.36) | 6.01 |
| Comoros | 858.86749 | 2,027.99 | 42.35 (40.20, 44.50) | 0.83 |
| Liberia | 1443.731 | 3,454.26 | 41.80 (40.15, 43.44) | 1.42 |
| Lesotho | 883.64641 | 2,336.67 | 37.82 (35.85, 39.78) | 0.99 |
| Madagascar | 3742.1846 | 8,749.88 | 42.77 (41.73, 43.81) | 3.58 |
| Mali | 2886.816 | 6,733.99 | 42.87 (41.69, 44.05) | 2.75 |
| Mauritania | 3251.296 | 7,226.04 | 44.99 (43.85, 46.14) | 2.92 |
| Malawi | 5610.464 | 12,402.99 | 45.23 (44.36, 46.11) | 5.01 |
| Mozambique | 3108.474 | 6,871.10 | 45.24 (44.06, 46.42) | 2.78 |
| Nigeria | 9387.0255 | 23,187.82 | 40.48 (39.85, 41.11) | 9.63 |
| Niger | 3132.79 | 7,918.97 | 39.56 (38.48, 40.64) | 3.31 |
| Namibia | 923.04854 | 2,331.67 | 39.59 (37.60, 41.57) | 0.98 |
| Rwanda | 2900.225 | 5,958.02 | 48.68 (47.41, 49.95) | 2.39 |
| Sierra Leone | 3293.202 | 7,775.86 | 42.35 (41.25, 43.45) | 3.19 |
| Senegal | 1975.602 | 4,011.92 | 49.24 (47.70, 50.79) | 1.61 |
| Chad | 4017.8798 | 11,083.16 | 36.25 (35.36, 37.15) | 4.80 |
| Togo | 2223.335 | 4,847.01 | 45.87 (44.47, 47.27) | 1.95 |
| Tanzania | 2779.0039 | 6,253.90 | 44.44 (43.20, 45.67) | 2.54 |
| Uganda | 3770.681 | 8,971.37 | 42.03 (41.01, 43.05) | 3.69 |
| South Africa | 739.23701 | 2,165.77 | 34.13 (32.14, 36.13) | 0.96 |
| Zambia | 2731.265 | 6,205.51 | 44.01 (42.78, 45.25) | 2.52 |
| Zimbabwe | 1846.128 | 4,709.47 | 39.20 (37.81, 40.59) | 1.98 |
| Pooled prevalence of single HRFB | | | 43.07 (42.88, 43.27) | 100.00 |

-55.4    0    55.4

**Fig 2. Pooled prevalence of single high-risk fertility behavior among married reproductive-age women across 35 SSA countries.**

## Discussion

This study aimed to estimate the prevalence of high-risk-fertility behaviors and associated factors among married women in 35 SSA countries. Accordingly, the pooled estimate of high-risk fertility behaviors was 77.7% [95% CI = 77.6%-77.9%], which is consistent with the studies conducted in Ethiopia (76.9%) [31] and (76.3%) [33]. However, this finding was higher than the studies in Kenya (70.8%) [37], Bangladesh (67.7%) [38] and (41.8%) [11], Ethiopia (62.1%) [41], Eastern Africa (57.6%) [27], and India (32%) [7]. Analysis of specific risky-fertility categories revealed that 43.1% [95% CI: 42.9%-43.3%] of women had experienced single fertility risk and this prevalence is lower than that of Bangladesh (45.6%) [38] but higher than the level reported for Eastern Africa (21.6%) [27] and India 26% [7]. This study also found that 31.4% [95% CI = 31.2%-31.6%] of women had multiple fertility risks. This finding is slightly lower than a study in Eastern Africa (36%) [27] but higher than other previous studies that reported

| Country | Multiple HRFB | Total | ES (95% CI) | Weight (%) |
|---|---|---|---|---|
| Angola | 2,596.33 | 6,586.05 | 39.42 (38.24, 40.60) | 2.33 |
| Burkina Faso | 3,223.47 | 10,658.91 | 30.24 (29.37, 31.11) | 4.27 |
| Benin | 2,526.67 | 8,861.71 | 28.51 (27.57, 29.45) | 3.68 |
| Burundi | 1,675.04 | 8,240.97 | 20.33 (19.46, 21.19) | 4.30 |
| DR Congo | 3,049.05 | 8,124.94 | 37.53 (36.47, 38.58) | 2.93 |
| Congo | 1,374.14 | 4,764.03 | 28.84 (27.56, 30.13) | 1.96 |
| Cote Divoire | 1,437.74 | 4,007.57 | 35.88 (34.39, 37.36) | 1.47 |
| Cameroon | 2,258.74 | 6,082.02 | 37.14 (35.92, 38.35) | 2.20 |
| Ethiopia | 3,413.30 | 7,885.36 | 43.29 (42.19, 44.38) | 2.72 |
| Gabon | 1,021.40 | 3,037.15 | 33.63 (31.95, 35.31) | 1.15 |
| Ghana | 1,013.95 | 4,185.91 | 24.22 (22.93, 25.52) | 1.93 |
| Gambia | 1,619.17 | 5,562.93 | 29.11 (27.91, 30.30) | 2.28 |
| Guinea | 2,000.20 | 5,862.19 | 34.12 (32.91, 35.33) | 2.21 |
| Kenya | 3,938.31 | 14,574.44 | 27.02 (26.30, 27.74) | 6.25 |
| Comoros | 752.54 | 2,027.99 | 37.11 (35.00, 39.21) | 0.73 |
| Liberia | 1,198.44 | 3,454.26 | 34.69 (33.11, 36.28) | 1.29 |
| Lesotho | 278.27 | 2,336.67 | 11.91 (10.60, 13.22) | 1.88 |
| Madagascar | 2,645.44 | 8,749.88 | 30.23 (29.27, 31.20) | 3.51 |
| Mali | 2,707.54 | 6,733.99 | 40.21 (39.04, 41.38) | 2.37 |
| Mauritania | 2,465.82 | 7,226.04 | 34.12 (33.03, 35.22) | 2.72 |
| Malawi | 3,760.49 | 12,402.99 | 30.32 (29.51, 31.13) | 4.97 |
| Mozambique | 2,366.15 | 6,871.10 | 34.44 (33.31, 35.56) | 2.57 |
| Nigeria | 9,014.48 | 23,187.82 | 38.88 (38.25, 39.50) | 8.25 |
| Niger | 3,836.14 | 7,918.97 | 48.44 (47.34, 49.54) | 2.68 |
| Namibia | 454.15 | 2,331.67 | 19.48 (17.87, 21.09) | 1.26 |
| Rwanda | 705.91 | 5,958.02 | 11.85 (11.03, 12.67) | 4.82 |
| Sierra Leone | 2,595.41 | 7,775.86 | 33.38 (32.33, 34.43) | 2.96 |
| Senegal | 985.04 | 4,011.92 | 24.55 (23.22, 25.88) | 1.83 |
| Chad | 5,942.07 | 11,083.16 | 53.61 (52.69, 54.54) | 3.77 |
| Togo | 1,230.03 | 4,847.01 | 25.38 (24.15, 26.60) | 2.16 |
| Tanzania | 1,816.70 | 6,253.90 | 29.05 (27.92, 30.17) | 2.57 |
| Uganda | 3,525.17 | 8,971.37 | 39.29 (38.28, 40.30) | 3.18 |
| South Africa | 242.77 | 2,165.77 | 11.21 (9.88, 12.54) | 1.84 |
| Zambia | 2,054.12 | 6,205.51 | 33.10 (31.93, 34.27) | 2.37 |
| Zimbabwe | 901.63 | 4,709.47 | 19.15 (18.02, 20.27) | 2.57 |
| Pooled prevalence of multiple HRFB | | | 31.37 (31.19, 31.55) | 100.00 |

-54.5    0    54.5

**Fig 3. Pooled prevalence of multiple high-risk fertility behaviors among married reproductive-age women across 35 SSA countries.**

a prevalence of (22.1%) [38] and (6%) [7]. The possible justifications for this discrepancy might be due to differences in the sample size, population characteristics, sociocultural contexts, and health system infrastructure across the study settings. The scope of the studies might also have contributed to this variation, in which the current study included 35 SSA countries, while the previous studies were limited to a specific country [31, 33], a single African region (i.e. Eastern Africa) [27] and 27 countries [36].

Additionally, our result showed disparities in the level of high-risk fertility practices across the included countries, ranging from 45.3% in South Africa to 89.9% in Chad. These variations might be attributed to differences in access to reproductive health services, utilization of fertility control methods, and health literacy across these countries. For instance, the high prevalence of risk fertility patterns in Chad could be due to low contraceptive uptake, where only 4% of adolescents use this service [42].

| Country | Overall HRFB | Total | ES (95% CI) | Weight (%) |
|---|---|---|---|---|
| Angola | 5,392.31 | 6,586.05 | 81.87 (80.94, 82.80) | 3.05 |
| Burkina Faso | 8,317.90 | 10,658.91 | 78.04 (77.25, 78.82) | 4.28 |
| Benin | 6,569.03 | 8,861.71 | 74.13 (73.22, 75.04) | 3.18 |
| Burundi | 6,148.04 | 8,240.97 | 74.60 (73.66, 75.54) | 2.99 |
| DR Congo | 6,631.04 | 8,124.94 | 81.61 (80.77, 82.46) | 3.72 |
| Congo | 3,417.17 | 4,764.03 | 71.73 (70.45, 73.01) | 1.62 |
| Cote Divoire | 3,090.76 | 4,007.57 | 77.12 (75.82, 78.42) | 1.56 |
| Cameroon | 4,784.36 | 6,082.02 | 78.66 (77.63, 79.69) | 2.49 |
| Ethiopia | 6,462.05 | 7,885.36 | 81.95 (81.10, 82.80) | 3.67 |
| Gabon | 2,336.14 | 3,037.15 | 76.92 (75.42, 78.42) | 1.18 |
| Ghana | 2,882.94 | 4,185.91 | 68.87 (67.47, 70.28) | 1.34 |
| Gambia | 4,159.03 | 5,562.93 | 74.76 (73.62, 75.90) | 2.03 |
| Guinea | 4,560.30 | 5,862.19 | 77.79 (76.73, 78.86) | 2.33 |
| Kenya | 9,995.81 | 14,574.44 | 68.58 (67.83, 69.34) | 4.65 |
| Comoros | 1,611.40 | 2,027.99 | 79.46 (77.70, 81.22) | 0.85 |
| Liberia | 2,642.17 | 3,454.26 | 76.49 (75.08, 77.90) | 1.32 |
| Lesotho | 1,161.91 | 2,336.67 | 49.73 (47.70, 51.75) | 0.64 |
| Madagascar | 6,387.62 | 8,749.88 | 73.00 (72.07, 73.93) | 3.05 |
| Mali | 5,594.36 | 6,733.99 | 83.08 (82.18, 83.97) | 3.29 |
| Mauritania | 5,717.12 | 7,226.04 | 79.12 (78.18, 80.06) | 3.01 |
| Malawi | 9,370.95 | 12,402.99 | 75.55 (74.80, 76.31) | 4.62 |
| Mozambique | 5,474.62 | 6,871.10 | 79.68 (78.72, 80.63) | 2.92 |
| Nigeria | 18,401.51 | 23,187.82 | 79.36 (78.84, 79.88) | 9.73 |
| Niger | 6,968.93 | 7,918.97 | 88.00 (87.29, 88.72) | 5.16 |
| Namibia | 1,377.20 | 2,331.67 | 59.07 (57.07, 61.06) | 0.66 |
| Rwanda | 3,606.14 | 5,958.02 | 60.53 (59.28, 61.77) | 1.71 |
| Sierra Leone | 5,888.61 | 7,775.86 | 75.73 (74.78, 76.68) | 2.91 |
| Senegal | 2,960.65 | 4,011.92 | 73.80 (72.44, 75.16) | 1.43 |
| Chad | 9,959.95 | 11,083.16 | 89.87 (89.30, 90.43) | 8.37 |
| Togo | 3,453.36 | 4,847.01 | 71.25 (69.97, 72.52) | 1.63 |
| Tanzania | 4,595.70 | 6,253.90 | 73.49 (72.39, 74.58) | 2.21 |
| Uganda | 7,295.85 | 8,971.37 | 81.32 (80.52, 82.13) | 4.06 |
| South Africa | 982.00 | 2,165.77 | 45.34 (43.25, 47.44) | 0.60 |
| Zambia | 4,785.38 | 6,205.51 | 77.12 (76.07, 78.16) | 2.42 |
| Zimbabwe | 2,747.76 | 4,709.47 | 58.35 (56.94, 59.75) | 1.33 |
| Overall prevalence of HRFB | | | 77.76 (77.60, 77.93) | 100.00 |

**Fig 4. Pooled prevalence of high-risk fertility behaviors among married reproductive-age women across 35 SSA countries.**

**Table 2. The result of random-effect logit models in predicting high-risk fertility behaviors among married reproductive-age women in SSA.**

| Parameters | Null Model | Model-II | Mode-III | Model-IV |
|---|---|---|---|---|
| Variance | 0.722 | 0.670 | 0.563 | 0.494 |
| Intraclass correlation coefficient | 0.180 | 0.169 | 0.146 | 0.131 |
| Proportion change in variance (PCV) | Reference | 0.072 | 0.220 | 0.316 |
| Median odds ratio (MOR) | 2.201 | 2.120 | 1.943 | 1.820 |
| **Model fitness** | | | | |
| Log-likelihood | -133753 | -131917 | -112620 | -112591 |
| Deviance | 267506 | 263834 | 225240 | 225182 |

**Table 3. Multivariable multilevel logistic regression analysis of individual and community-level determinants of high-risk fertility behaviors among married women in sub-Saharan African countries.**

| | Adjusted Analysis (AOR (95% CI)) | | |
|---|---|---|---|
| | **Model II** | **Model III** | **Model IV** |
| **Individual-level factors** | | | |
| **Age at first marriage** | | | |
| ≥18 year | 1.00 | - | 1.00 |
| <18 year | 3.54 (3.34, 3.75) | | 3.70 (3.59, 3.81)* |
| **Nature of marriage** | | | |
| Monogamy | 1.00 | - | 1.00 |
| Polygamy | 1.33 (1.27, 1.40) | | 1.32 (1.28, 1.37)* |
| **Woman's education** | | | |
| Higher education | 1.00 | - | 1.00 |
| Primary education | 1.76 (1.65, 1.86) | | 1.67 (1.61, 1.73)* |
| No formal education | 2.34 (2.13, 2.58) | | 2.14 (2.06, 2.23)* |
| **Partner/husband education** | | | |
| Higher education | 1.00 | | 1.00 |
| Primary education | 1.26 (1.18, 1.34) | - | 1.23 (1.11, 1.27)* |
| No formal education | 1.58 (1.47, 1.69) | | 1.50 (1.44, 1.62)* |
| **Head of household** | | | |
| Female | 1.00 | - | 1.00 |
| Male | 1.13 (1.08, 1.19) | | 1.14 (1.11, 1.18)* |
| **Household wealth index** | | | |
| Rich | 1.00 | - | 1.00 |
| Middle | 1.26 (1.16, 1.36) | | 1.28 (1.23, 1.33)* |
| Poor | 1.27 (1.12, 1.43) | | 1.27 (1.22, 1.32)* |
| **Unmet need for FP** | | | |
| No | 1.00 | - | 1.00 |
| Yes | 1.38 (1.31, 1.45) | | 1.38 (1.34, 1.43)* |
| **Couple's fertility preference** | | | |
| Concordant | 1.00 | - | 1.00 |
| Discordant | 1.10 (1.05, 1.15) | | 1.15 (1.12, 1.19)* |
| **Community-level factors** | | | |
| **Residence** | | | |
| Urban | - | 1.00 | 1.00 |
| Rural | | 1.92 (1.78, 2.07) | 1.86 (1.81, 1.93)* |
| **Community-level women illiteracy** | | | |
| High | - | 1.00 | 1.00 |
| Low | | 1.05 (0.89, 1.23) | 0.99 (0.94, 1.10) |
| **Community early marriage practice** | | | |
| Low | - | 1.00 | 1.00 |
| High | | 1.58 (1.48, 1.69) | 1.37 (1.42, 1.49)* |
| **Community-level women empowerment** | | | |
| High | - | 1.00 | 1.00 |
| Low | | 1.25 (1.22, 1.29) | 1.24 (1.22, 1.27)* |

HRFB: high-risk fertility behavior; FP: family planning; SSA: sub-Saharan Africa. *Significant at p-value less than 0.05.

This study found that women who married before the age of 18 were more than three times more likely to engage in high-risk fertility behavior than women who married at or after the age of 18. This finding is supported by a previous study in Nigeria that reported a significant relationship between child marriage and high-risk births [43]. The possible explanation for this finding is that women who marry at younger ages are more likely to have higher lifetime fertility due to early and repeated childbirths [44, 45] and have limited decision-making autonomy over the use of fertility control methods [46, 47].

In this study, women in polygamous unions had higher odds of being in a high-risk fertility category than those in monogamous relationships. This result is in line with the study findings from sub-Saharan Africa [36]. The negative effect of polygamous marriage on maternal contraceptive use which in turn impacts suboptimal birth spacing and high parity births might have contributed to this finding [48].

Maternal education was also found as a factor that influenced the occurrence of high-risk fertility patterns. For instance, women with no formal and primary education were more likely to engage in risky fertility behavior than those who attended higher education. This finding is consistent with the results of the previous studies [27, 31, 36, 38]. This could be explained by the fact that uneducated women are likely to have poor knowledge and awareness of sexual and reproductive health services which negatively affects their decision to use these services [49].

Consistent with studies in Africa [27, 31, 36] and Bangladesh [38], this study also found that women whose partners had no formal and primary-level education were at an increased risk of experiencing high-risk fertility behaviors compared to women whose partners have completed higher education. This could be because uneducated husbands have a lesser understanding of fertility control methods and thus refuse their spouse to use contraception which subsequently forces the women to have high-risk pregnancies [49].

This study also found that women from male-headed households were 14% more likely to experience high-risky fertility behaviors than women from female-headed households. This is consistent with a study conducted in Eastern Africa, which found a lower likelihood of risky fertility behaviors among women living in female-headed families [27].

It was also revealed that women with unmet family planning needs were 38% more likely to experience high-risk fertility behaviors than women with met needs. This could be because an unsatisfied and unmet risk-based need for modern contraception forces women to rely on traditional methods that are less effective and therefore increase their exposure to risky fertility patterns like closely spaced births and unintended pregnancies [50].

A couple's fertility preference was also found to have a significant influence on maternal fertility behaviors. Women whose partners prefer to have more or fewer children had higher odds of having risky fertility behavior compared to women whose husbands have a concordant fertility preference. This could be because although women intend to limit or space their childbearing, deep-rooted gender norms and husbands' dominance in decisions regarding the number of children limit the women's tendency to use fertility control methods to avoid unwanted pregnancies.

According to this study, the odds of risky fertility behavior were higher among women who resided in rural settings compared to those who lived in urban areas. Similarly, a previous study in Eastern Africa reported higher odds of risky pregnancy behaviors among rural residents [27]. This might be because rural women are less likely to be educated, have limited access to health information and reproductive health services, and thus tend to engage in high-risk fertility behaviors than those in urban settings [51, 52].

Community-level early marriage practice was also found to have a significant effect on high-risk fertility behaviors. Women who lived in the communities with high childhood

marriage practice had a 37% increased odds of experiencing high-risk fertility behavior compared to those in the communities where this practice was low. The possible explanation for this finding is that in the communities where early adolescence and childhood marriage are common, women's acceptance of such cultural practices increases their vulnerability to high fertility and low utilization of maternal health services like contraception that directly affects their fertility behaviors [53].

Furthermore, this study showed that the likelihood of engaging in risky pregnancy behaviors was higher among women from communities with low women empowerment compared to women in the areas with high women empowerment. This might be because underpowered women have a limited opportunity to access health services and lower participation in decisions regarding contraceptive use [54, 55].

## Policy implications

Our findings underscore the need to design and implement contextualized interventions, particularly in settings with a high level of risky fertility behaviors to address the specific risk factors identified in the study. Moreover, strengthening and revisiting the existing sexual and reproductive health programs, increasing access to family planning services, and supporting enforcement of laws and policies governing early and polygamous marriages are critical for reducing risky reproductive behaviors.

## Strengths and limitations

The study used data from a nationally representative DHS of 35 countries that used a validated tool, and a larger sample size. It also employed an advanced statistical technique (i.e multilevel modeling) to account for the nested or hierarchical nature of the DHS data. However, it is impossible to demonstrate the cause-and-effect relationship between the explanatory and outcome variables due to the cross-sectional nature of the data. In addition, there might be a recall bias since participants were asked about the events that took place five years or more preceding the survey.

## Conclusions

Over three-quarters of married reproductive-age women in SSA were engaged in high-risk fertility behaviors with huge variation across the countries. Both individual and community-level factors were found to have a significant effect on these fertility risks. Therefore, addressing the modifiable risk factors like improving access to need-based contraceptive methods and empowering couples through education for a better understanding of their reproductive health with particular attention to rural settings are important in reducing these fertility behaviors. Our finding also suggests the need to strengthen the policies regulating the prohibition of early and polygamous marriages and look into context-specific interventions in the areas with high prevalence. Most importantly, exploring the specific cultural, social, and economic factors contributing to these fertility practices and conducting qualitative studies to better understand the context and drivers behind these fertility behaviors is also important.

## Acknowledgments

The authors thank ICF International for granting access to the dataset used in this study.

## Author Contributions

**Conceptualization:** Kusse Urmale Mare, Setognal Birara Aychiluhm, Getahun Fentaw Mulaw, Kebede Gemeda Sabo, Mekuriyaw Gashaw Asmare, Betel Zelalem Wubshet, Tsion Mulat Tebeje, Beminate Lemma Seifu.

**Data curation:** Kusse Urmale Mare.

**Formal analysis:** Kusse Urmale Mare, Setognal Birara Aychiluhm, Getahun Fentaw Mulaw, Kebede Gemeda Sabo, Betel Zelalem Wubshet, Tsion Mulat Tebeje, Beminate Lemma Seifu.

**Investigation:** Kusse Urmale Mare.

**Methodology:** Kusse Urmale Mare, Setognal Birara Aychiluhm, Getahun Fentaw Mulaw, Kebede Gemeda Sabo, Mekuriyaw Gashaw Asmare, Betel Zelalem Wubshet, Tsion Mulat Tebeje, Beminate Lemma Seifu.

**Software:** Kusse Urmale Mare, Setognal Birara Aychiluhm, Getahun Fentaw Mulaw, Kebede Gemeda Sabo, Mekuriyaw Gashaw Asmare, Tsion Mulat Tebeje, Beminate Lemma Seifu.

**Validation:** Kusse Urmale Mare.

**Visualization:** Kusse Urmale Mare, Kebede Gemeda Sabo, Mekuriyaw Gashaw Asmare.

**Writing – original draft:** Kusse Urmale Mare, Setognal Birara Aychiluhm, Getahun Fentaw Mulaw, Kebede Gemeda Sabo, Mekuriyaw Gashaw Asmare, Betel Zelalem Wubshet, Tsion Mulat Tebeje, Beminate Lemma Seifu.

**Writing – review & editing:** Kusse Urmale Mare, Setognal Birara Aychiluhm, Getahun Fentaw Mulaw, Kebede Gemeda Sabo, Mekuriyaw Gashaw Asmare, Betel Zelalem Wubshet, Tsion Mulat Tebeje, Beminate Lemma Seifu.

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
