## [Decision Letter · Decision Letter 0]

5 Jun 2024

PGPH-D-24-00717

High-risk fertility behaviors and associated factors among married reproductive-age women in sub-Saharan Africa: a multilevel mixed-effect analysis of nationally representative data from 35 countries

Dear Mr.Kusse Urmale Mare,

Thank you for submitting your manuscript to PLOS Global Public Health. After careful consideration, we feel that it has merit but does not fully meet PLOS Global Public Health’s publication criteria as it currently stands. Therefore, we invite you to submit a revised version of the manuscript that addresses the points raised during the review process.

We look forward to receiving your revised manuscript.

Kind regards,

Jayanta Kumar Bora,PhD

Academic Editor

Journal Requirements:

1. Please provide separate figure files in .tif or .eps format only and remove any figures embedded in your manuscript file. Please also ensure all files are under our size limit of 10MB.

2. We noticed that you used "unpublished" in the manuscript. We do not allow these references, as the PLOS data access policy requires that all data be either published with the manuscript or made available in a publicly accessible database. Please amend the supplementary material to include the referenced data or remove the references.

Additional Editor Comments (if provided):

Reviewers' comments:

Reviewer's Responses to Questions

**Comments to the Author**

1. Does this manuscript meet PLOS Global Public Health’s publication criteria? Is the manuscript technically sound, and do the data support the conclusions? The manuscript must describe methodologically and ethically rigorous research with conclusions that are appropriately drawn based on the data presented.

Reviewer #1: Yes

Reviewer #2: Yes

Reviewer #3: Yes

2. Has the statistical analysis been performed appropriately and rigorously?

Reviewer #1: Yes

Reviewer #2: Yes

Reviewer #3: Yes

3. Have the authors made all data underlying the findings in their manuscript fully available (please refer to the Data Availability Statement at the start of the manuscript PDF file)?

Reviewer #1: Yes

Reviewer #2: Yes

Reviewer #3: Yes

4. Is the manuscript presented in an intelligible fashion and written in standard English?

Reviewer #1: Yes

Reviewer #2: Yes

Reviewer #3: Yes

5. Review Comments to the Author

Reviewer #1: Overall, the article provides a comprehensive analysis of high-risk fertility behaviors among married women in sub-Saharan African countries, utilizing data from 35 countries and employing advanced statistical techniques. This study is unique enough and more convincing timely output of the topic covered. The study sheds light on both individual and community-level factors associated with the risk of fertility, offering valuable insights for policymakers and public health practitioners.

However, there are several points that could enhance the clarity, robustness, and impact of the research:

Comment 1: Methods (page 6): the methodology section seems to be strong enough however, it outlines the use of multivariable multilevel logistic regression to analyze determinants of high-risk fertility behaviors, more detailed information on model selection criteria and potential confounders could strengthen the methodological rigor. Providing justification for the choice of specific variables and discussing potential biases would further enhance the validity of the findings.

Comment 2: Methods (page 6): there has been a potential sampling bias as given the cross-sectional nature of the data and the possibility of recall bias. Providing information on the representativeness of the sample and efforts taken to ensure data reliability could support the credibility of the study findings. Also, what mechanism did you take to minimize the bias is to be reported.

Comment 3: Results (page 10): the article presents a wealth of data on high-risk fertility behaviors across various sociodemographic and reproductive characteristics. However, a deeper exploration of the implications of these findings for different population (as categorized) subgroups would enrich the discussion. For instance, discussing the differential impact of interventions based on age, education, and geographic sub-location or based on the outcome variable could provide valuable insights for targeted public health strategies.

Comment 4: the authors can add a section or sub-section under the discussion in the name of Policy Implications. While the discussion touches upon the importance of addressing modifiable risk factors and strengthening policies, further elaboration on specific policy recommendations and their feasibility within the context of sub-Saharan African countries would be beneficial. Providing concrete suggestions for policymakers, such as investments in education and family planning services or interventions at the community level, could enhance the practical relevance of the study.

Comment 5: you can provide an outline on future research directions guided by the study findings and implications. Considering the complexity of factors influencing high-risk fertility behaviors, it would be valuable to propose avenues for future research. Exploring emerging trends, such as the impact of digital health interventions or cultural shifts, could offer new perspectives on addressing fertility-related challenges in the region, or whatever the authors think of appropriate based on the study outcome.

Reviewer #2: Authors have studied “High-risk fertility behaviors and associated factors among married reproductive-age women in sub-Saharan Africa: a multilevel mixed-effect analysis of nationally representative data from 35 countries”. The present work is nice piece of work. I can give the recommendation for publication but before this, some minor revisions need to be addressed.

1) What is advantage of this study? write this in introduction section and in conclusion.

2) The research gap should be clearly written.

3) Cite the properly the used equations.

4) Improve the discussion section for the graphs presented in the current work.

5) Please some related new work published in this journal.

6) Improve the conclusion and abstract.

7) What are the limitations of the present work and what are the future recommendations. Please mention them in abstract.

8) Check the entire manuscript regarding grammar and spelling mistakes.

Reviewer #3: In the paper, the authors have analyzed pooled prevalence of high-risk fertility behaviors and associated factors among married women in sub Saharan African countries.

I found this paper quite interesting and well managed. To further improve this study, I suggest the following:

Authors should also include "Religion" as a explanatory variable as, for this study, it can play can important role and make the study more comprehensive.

6. PLOS authors have the option to publish the peer review history of their article (what does this mean?). If published, this will include your full peer review and any attached files.

**Do you want your identity to be public for this peer review?** For information about this choice, including consent withdrawal, please see our Privacy Policy.

Reviewer #1: **Yes: **Md Al Amin

Reviewer #2: No

Reviewer #3: No

---

## [Decision Letter · Decision Letter 1]

9 Aug 2024

High-risk fertility behaviors and associated factors among married reproductive-age women in sub-Saharan Africa: a multilevel mixed-effect analysis of nationally representative data from 35 countries

PGPH-D-24-00717R1

Dear Mr. Mare,

We are pleased to inform you that your manuscript 'High-risk fertility behaviors and associated factors among married reproductive-age women in sub-Saharan Africa: a multilevel mixed-effect analysis of nationally representative data from 35 countries' has been provisionally accepted for publication in PLOS Global Public Health.

Best regards,

Julia Robinson

Executive Editor

Reviewer Comments (if any, and for reference):

Reviewer's Responses to Questions

**Comments to the Author**

1. If the authors have adequately addressed your comments raised in a previous round of review and you feel that this manuscript is now acceptable for publication, you may indicate that here to bypass the “Comments to the Author” section, enter your conflict of interest statement in the “Confidential to Editor” section, and submit your "Accept" recommendation.

Reviewer #1: All comments have been addressed

Reviewer #2: All comments have been addressed

2. Does this manuscript meet PLOS Global Public Health’s publication criteria? Is the manuscript technically sound, and do the data support the conclusions? The manuscript must describe methodologically and ethically rigorous research with conclusions that are appropriately drawn based on the data presented.

Reviewer #1: Yes

Reviewer #2: Yes

3. Has the statistical analysis been performed appropriately and rigorously?

Reviewer #1: Yes

Reviewer #2: Yes

4. Have the authors made all data underlying the findings in their manuscript fully available (please refer to the Data Availability Statement at the start of the manuscript PDF file)?

Reviewer #1: Yes

Reviewer #2: Yes

5. Is the manuscript presented in an intelligible fashion and written in standard English?

Reviewer #1: No

Reviewer #2: Yes

6. Review Comments to the Author

Reviewer #1: The authors have perfectly addressed all the comments. The manuscript is now up to the marks, but needs language editing for proper publishable English.

Thank you for the efforts of the authors enhance the scholarly work.

Reviewer #2: Accepted in its current revised form.

7. PLOS authors have the option to publish the peer review history of their article (what does this mean?). If published, this will include your full peer review and any attached files.

**Do you want your identity to be public for this peer review?** For information about this choice, including consent withdrawal, please see our Privacy Policy.

Reviewer #1: **Yes: **Md Al Amin

Reviewer #2: No
